# Retention among transgender women treated with dolutegravir associated with tenofovir/lamivudine or emtricitabine in Argentina: TransViiV study

Claudia E. Frola[1,2]*, Inés Aristegui[1,3], María I. Figueroa[1], Pablo D. Radusky[1,4], Nadir Cardozo[1,5], Virginia Zalazar[1], Carina Cesar[1], Patricia Patterson[1], Valeria Fink[1], Ana Gun[1], Pedro Cahn[1], Omar Sued[1]

1 Research Department, Fundación Huésped, Buenos Aires, Argentina, 2 Division of Infectious Diseases, Juan A. Fernández Hospital, Buenos Aires, Argentina, 3 Department of Research in Psychology, Universidad de Palermo, Buenos Aires, Argentina, 4 Faculty of Psychology, Universidad de Buenos Aires, Buenos Aires, Argentina, 5 Association of Transvestites, Transsexuals, and Transgenders of Argentina (A.T.T.T.A.), Buenos Aires, Argentina

* claudia.frola@huesped.org.ar

**Data Availability Statement:** All relevant data are within the manuscript. We can provide access without limitations to all the dataset.

## Abstract

In Argentina, transgender women (TGW) have a high HIV prevalence (34%). However, this population shows lower levels of adherence, retention in HIV care and viral suppression than cisgender patients. The World Health Organization (WHO) recommends the transition to dolutegravir (DTG)-based regimens to reduce adverse events and improve adherence and retention. The purpose of this study was to determine retention, adherence and viral suppression in naïve TGW starting a DTG-based first-line antiretroviral treatment (ART) and to identify clinical and psychosocial factors associated with retention. We designed a prospective, open-label, single-arm trial among ART-naïve HIV positive TGW (Clinical Trial Number: NCT03033836). Participants were followed at weeks 4, 8, 12, 24, 36 and 48, in a trans-affirmative HIV care service that included peer navigators, between December, 2015 and May, 2019. Retention was defined as the proportion of TGW retained at week 48 and adherence was self-reported. Viral suppression at <50 copies/mL was evaluated using snapshot algorithm and as per protocol analysis. Of 75 TGW screened, 61 were enrolled. At baseline, median age was 28 y/o., HIV-1-RNA (pVL) 46,908 copies/mL and CD4+ T-cell count 383 cells/mm$^3$. At week 48, 77% were retained and 72% had viral suppression (97% per protocol). The regimen was well tolerated and participants reported high adherence (about 95%). Eleven of the fourteen TGW who discontinued or were lost to follow-up had undetectable pVL at their last visit. Older age was associated with better retention. DTG-based treatment delivered by a trans-competent team in a trans-affirmative service was safe and well tolerated by TGW and associated with high retention, high adherence and high viral suppression at 48 weeks among those being retained.

**Funding:** The present study was funded by an Investigator Initiated Research Grant from ViiV Healthcare, GSK202037. The funders had no role in study design, data collection and analysis, decision to publish, or preparation of the manuscript.

**Competing interests:** Co-author Omar Sued is a PLOS ONE Editorial Board member. However, does not alter the authors' adherence to all the PLOS ONE policies on sharing data and materials.

## Introduction

Transgender women (TGW) remain a key population within the global HIV pandemic, with estimates that suggest that their risk of acquiring HIV is almost 50 times higher than the rest of the population [1]. Argentina has a concentrated HIV epidemic among specific groups, with a prevalence estimated at 34% in TGW, compared to 0.4% in the general population [2]. In contrast, this population consistently shows low levels of adherence to antiretroviral treatment (ART) and retention in HIV care, in comparison with cisgender patients, and therefore, TGW are less likely to achieve viral suppression [3–6]. At the root of this adverse outcome, several factors can be identified, such as experiences of gender identity-related stigma and discrimination, particularly within the healthcare system, and high levels of psychosocial and economic vulnerability (e.g., high prevalence of mental health problems and substance use, barriers to access formal employment, among others) [5, 7–11].

Treatment-related factors may also affect the level of adherence to antiretroviral treatment (ART). Complexity of regimen (dosing frequency, pill burden) and presence of side effects are frequent correlates of treatment abandonment [12]. Transition to integrase inhibitors (INIs) based regimens is associated with lower discontinuation rate and higher efficacy to increase viral suppression and, potentially, higher retention in HIV care [13]. For this reason, currently the World Health Organization (WHO) [14] recommends dolutegravir (DTG) with two nucleoside/nucleotide reverse transcriptase inhibitors (NRTIs) as the preferred ART for naïve individuals based on its better efficacy and safety profile, in order to reduce adverse events, improve adherence and retention in HIV care and as a response to increments of the primary resistance to non-nucleoside reverse transcriptase inhibitors. As a possible disadvantage, only an increased risk of weight gain in the short term has been reported in those starting DTG-based regimens [15].

Increasing TGW's ART adherence and retention is critical for this population's quality of life and for public health. Evidence demonstrates that provision of HIV treatment, in addition to preventing disease progression and death, greatly reduces transmission to sex partners, an approach known as treatment as prevention (TasP) [16]. Furthermore, retention is fundamental for the timely identification of side effects, treatment abandonment or plasma viral load (pVL) rebound in order to adjust ART to ensure viral suppression [17–19]. The scarce information about retention in HIV care among TGW reinforces the need to study all the steps of the HIV cascade in this population [20], especially regarding the use of DTG-based regimens.

We, therefore, hypothesized that a DTG-based treatment would be an effective regimen in ART-naïve, HIV-1-infected TGW. We conducted this 48-week study to determine retention, adherence and viral suppression of TGW treated with DTG plus tenofovir disoproxil fumarate (TDF) coformulated with either emtricitabine (FTC) or lamivudine (3TC) (DTG + TDF-XTC) and to identify clinical and psychosocial factors associated with retention.

## Methods

### Study setting

This study was conducted in a non-governmental organization that provides free-of-charge trans-competent and trans-affirmative HIV care in the context of clinical research in Buenos Aires, Argentina. Our trans-affirmative healthcare service includes a) use of patients' preferred name and pronoun in interactions, clinical records and forms (which include sex assigned at birth and gender identity); b) an interdisciplinary trans-competent trained staff, aware of transgender people's needs and accepting of their identities; c) integration of multiple services for this community (e.g., HIV, gender-affirming medical procedures, anal health) to simplify

service delivery; d) adjustment to transgender populations' social contexts (e.g., flexible scheduling and hours); and e) inclusion of transgender peer navigators. Peer navigators function as a bridge between the research site and the transgender community. Some of their main tasks are: a) to provide health information to their peers adapting it to their community and making it more accessible and comprehensible, b) to invite potential participants and to enroll them in the studies, c) to verify that they understood the informed consent correctly and to answer any concern about it, d) to assist transgender people in obtaining medical appointments and in navigating the healthcare service, e) to remind participants their upcoming visits, and f) to contact lost to follow-up participants to re-engage them in the study.

## Study design

This was a prospective, open-label, single-arm trial of DTG-TDF/XTC. This study was designed in a national context of use of efavirenz-containing triple antiretroviral regimen with possible changes in sleep quality. This regimen could particularly affect a population with high proportions of engagement in nightly sex work, such as TGW [11], therefore negatively impacting adherence and retention. Eligible participants were ART-naïve, HIV-1-infected, self-identified as TGW, $\geq$18 years old, without evidence of genotypic viral resistance to 3TC, FTC or TDF as per IAS-USA 2013 resistance panel [21]. Exclusion criteria included severe hepatic impairment (Child-Pugh B or C), hepatitis C requiring treatment, creatinine clearance <50 ml/min and plans of moving out of the city within the next year. Study participants received one daily pill of open-label DTG 50 mg plus either one daily pill of TDF 300 mg /3TC 300 mg or /FTC 200 mg. Treatment was provided on site at the following visits: baseline and weeks 4, 8, 12, 24, 36 y 48.

Participants were recruited by outreach efforts of peer navigators, through testing campaigns conducted in places were transgender people gather or live, and through collaboration with a local transgender community-based organization. Eligible participants started ART and were followed for 48 weeks according to study protocol and procedures, between December, 2015 and May, 2019 (Table 1). Participants with significant suicidal ideation were assessed for suicide risk by a mental healthcare provider, and referral to mental health services was facilitated to those who required it.

According to local regulations, the registration of this study in the national registry of clinical trials (ReNIS—Registro Nacional de Investigaciones en Salud) was not required. However, we decided to register it as we were aware that it was a requirement for the publication of the study's results. Consequently, it was registered after the enrollment of participants started, under Clinical Trial Number NCT03033836. The authors confirm that all ongoing and related trials for this drug/intervention are registered.

**Measures.** Main variables for this study were measured as follows:

- *Retention*: this primary endpoint was defined as the proportion of TGW retained in the study at weeks 24 and 48.

- *Adherence to ART*: self-reported medication adherence was assessed using the Adult AIDS Clinical Trials Group Adherence Questionnaire [22]. A visual analogue scale was also used, composed of a line with a range of possible quantitative scores (0–100%) and the TGWs were instructed to place a mark on a point on it that described their uptake of ART in the last four weeks. An average percentage calculation of the adherence of each participant was performed at each study visit, including weeks 24 and 48.

- *Viral suppression*: proportion of individuals with pVL<50 copies/ml at week 48 using the FDA snapshot algorithm (missing, switch or discontinuation = failure) for the intention-to-

**Table 1. Schedule of procedures.**

| Procedures | SCR | BSL | Week 4 | Week 8 | Week 12 | Week 24 | Week 36 | Week 48 | Final Visit |
|---|---|---|---|---|---|---|---|---|---|
| **Informed Consent** | X | | | | | | | | |
| **Medical History** | X | | | | | | | | |
| **Physical Exam** | X | X | X | X | X | X | X | X | X |
| **Vital Signs** | X | X | X | X | X | X | X | X | X |
| **Electrocardiogram** | X | | | | | | | | |
| **Concomitant Drugs** | X | X | X | X | X | X | X | X | |
| **Psychosocial Questionnaires[a]** | | X | | | | X | | X | |
| **AEs and Adherence** | | X | X | X | X | X | X | X | X |
| **HIV Genotype** | X | | | | | | | | |
| **Serologies[b]** | X | | | | | | | | |
| **HIV-1 RNA** | X | X | | X | X | X | X | X | |
| **CD4 Cell Count** | X | X | | | X | X | X | X | |
| **Complete Blood Count[c]** | X | X | X | | X | X | X | X | |
| **Basic Chemistry[d]** | X | | X | | X | | X | | |
| **Complete Chemistry[e]** | | X | | | | X | | X | |

**Abbreviations:** SCR, screening; BSL, basal

[a] Socio-demographic questionnaire, suicidal ideation, AUDIT, DAST-10.

[b] HBV anti-core, HBVsAg, HCV IgG, VDRL.

[c] Hemoglobin, hematocrit, erythrocytes, leukocytes, thrombocytes, neutrophils, basophils, lymphocytes, monocytes, reticulocytes, MCV, MCH, MCHC, platelets, prothrombin time, activated thromboplastin time.

[d] ALT, AST, total bilirubin, creatinine.

[e] ALT, AST, alkaline phosphatase, amylase, gamma-GT, sodium, potassium, calcium, chloride, bicarbonate, total bilirubin, direct bilirubin, indirect bilirubin, lactate, creatinine, urea, total cholesterol, HDL cholesterol, LDL cholesterol, triglycerides, lactate dehydrogenase, total protein, albumin, urine dipstick.

treat (ITT) exposed population. Viral genotype was analyzed (TRUGENE® HIV-1 Genotyping Kit) at the screening visit and at the time of protocol-defined virological failure (PDVF), defined as pVL $\geq$1,000 copies/ml at week 24 or 36 or $\geq$50 copies/ml at week 48; or a confirmed viral rebound ($>$200 copies/ml) after pVL $<$50 copies/ml. Participants were required to withdraw from the study if PDVF was confirmed.

- *Safety and tolerability*: it was assessed considering the frequency, type and severity of adverse events (AEs), serious adverse events (SAEs) and laboratory abnormalities graded according to the Division of AIDS (DAIDS) Table for Grading the Severity of Adults and Pediatric Adverse Events, version 2.0- November 2014. Likewise, its possible relationship with the study regimen was evaluated.

- *Clinical characteristics*: these variables included a) immunological status related to HIV (CD4 and pVL count); b) syphilis basal prevalence and incidence along the study; c) co-infection with hepatitis B and/or C; d) concomitant medication (e.g., hormone therapy) at baseline, and e) presence of overweight or obesity according to the body mass index corresponding to $\geq$25 or $\geq$30, respectively.

- *Psychosocial variables*: these variables were measured at baseline, 24 and 48 weeks with a questionnaire that included: a) sociodemographic variables (educational level, current engagement in sex work, housing stability, migration); b) 4-item suicidal ideation screener; c) Alcohol Use Disorders Identification Test (AUDIT, 10 items, scores $\geq$ 8 indicate hazardous alcohol use), and d) Drug Abuse Screening Test (DAST-10, 10 items, scores $\geq$ 6 indicate possible drug abuse or dependence problems) [23–25].

## Statistical analyses

Sample size was not determined in advance and a convenience sampling was conducted. All data were anonymized and analyzed using the Statistical Package for the Social Sciences (SPSS) 24 software.

Descriptive statistics were presented for the primary and secondary outcomes, with median and interquartile ranges (IQRs) or frequencies and proportions (%), as appropriate. Comparisons between "retained" and "not retained" individuals were performed to identify baseline clinical and sociodemographic factors associated with loss of follow-up at week 48, using Chi-square/Fisher's exact test for categorical variables, and Mann-Whitney's U test for continuous variables when normality was not confirmed. Longitudinal differences between baseline and week 48 were assessed for CD4 count and weight, using Wilcoxon rank-sum test, as these variables showed non-normal distributions. A survival analysis of retained participants was performed with R (survival package). Time to event was calculated by Kaplan-Meier method. For all the analyses, a $p$ value $< .05$ was considered significant. However, alpha levels below 10% were considered as indicating a trend.

## Ethics statement

TransViiV study was evaluated and approved by the institutional review board under the number FH-17. The study was carried out following the good clinical practices. All individuals provided a written informed consent before participation in any study procedure. Participation was voluntary. At each visit, participants received a $150 Argentine pesos compensation (approximately, 15 USD at the moment of the study) to cover transportation costs, and a coupon exchangeable for a basic breakfast or meal.

# Results

Between December, 2015 and May, 2018, 75 TGW were screened. Of these, 61 were enrolled and received the study treatment. Fourteen TGW (18%) were considered screening failure, being the most frequent reasons non-amplification of the genotypic test and plans to move to another city in the following year (Fig 1).

## Baseline clinical and psychosocial characteristics

At enrollment, most patients were asymptomatic, with 96% CDC staged as class A. Main HIV route of transmission was unprotected sexual contact. Within the baseline clinical characteristics (Table 2), 20% of the TGW had a CD4 count $<200$ cells/mm$^3$ and 40% had high pVL ($>100,000$ c/mL). With respect to other sexually transmitted infections, the baseline prevalence and incidence of syphilis was 36% (n = 22) and 29.5%, respectively. HIV co-infection with hepatitis B or C was $\leq$5%.

Baseline psychosocial characteristics show high levels of social vulnerability (e.g., 53% [n = 32] reported unstable housing; 77% [n = 47], current engagement in sex work; 61% [n = 37], incomplete high school education). Regarding mental health indicators, 26% (n = 16) of these TGW reported significant suicidal ideation in the last 2 weeks. Moreover, 66% reported drug use in the last year (53% [n = 32] reported cocaine use), 13% [n = 8] evidenced possible drug abuse and 53% (n = 32) presented hazardous alcohol use.

## Adherence, retention and viral suppression

Regarding adherence, the mean for week 24 was 86.9% and the median value was 95% (76.6–95). For week 48, the mean reported adherence was 86.3% and the median value was also 95% (76.6–95).

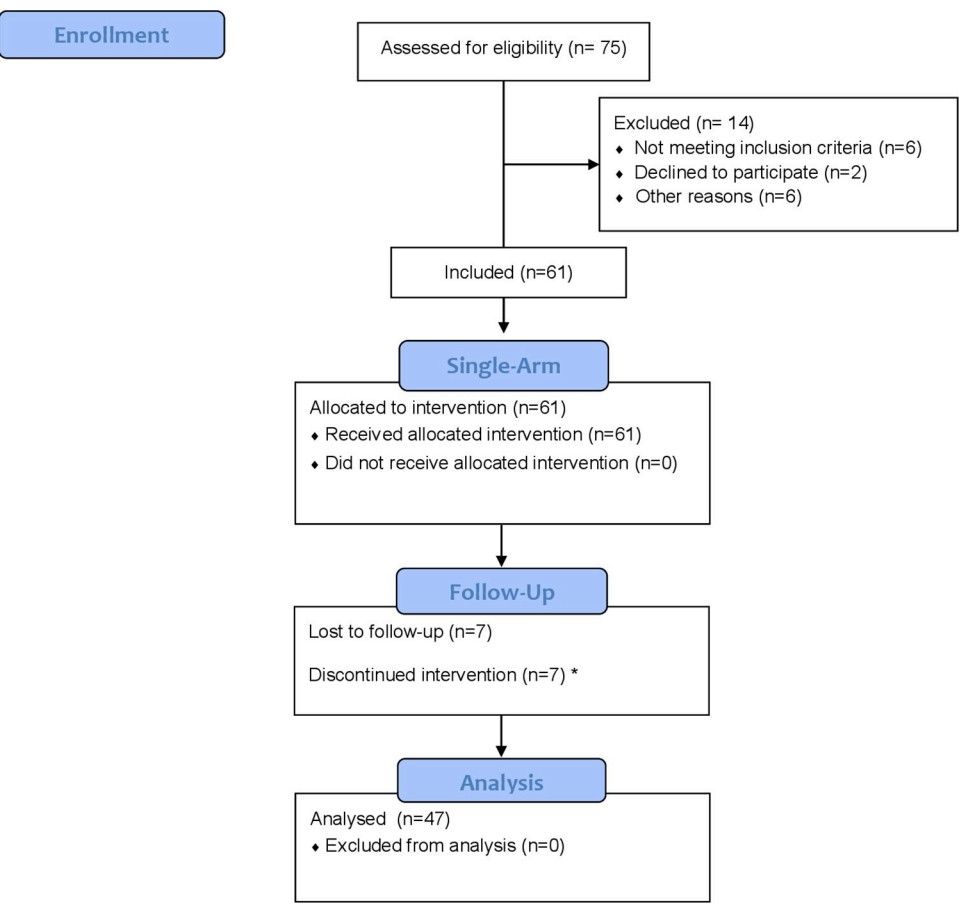

**Fig 1. Screening to week 48.** NOTE. [*] hyperamylasemia related to a prior history of alcohol abuse. [#] false-positive HIV test. LTFU, lost to follow-up. SAEs: suicide attempt (1) and death (1).

Regarding retention, 82% were retained in the study at week 24 and this proportion decreased to 77% at week 48. Of the 47 retained participants, 2 did not have a pVL at week 48 but, as they attended the clinic, they were considered as retained for the analyses. Although 14 (22%) participants were discontinued or lost to follow-up (Fig 1); 79% of them had undetectable pVL at their last visit. Of the retained TGW who performed the laboratory test at 48 weeks, 72% had viral suppression using snapshot algorithm ITT-exposed and 97% (44/45) did so in the per-protocol analysis. One patient had PDVF at week 48, but the pVL was <200 copies/ml and it could not be amplified for genotypic analysis.

In relation to the CD4 count, a statistically significant increase was recorded between baseline and week 48 ($z$ = -4.679, $p$ = 0.000). The median increase in CD4 count between baseline and week 48 was 276 (median CD4 count at baseline was 389 [216–614]; median at week 48 was 665 [493–815]). This is an increase of 71% on the median value for CD4 count.

## Survival analysis

The starting point for this analysis is baseline visit when participants received the first treatment medication. Time between this visit and the next medication dispensed (next visit date) was calculated until the first event of no medication dispensed occurred. When this happened, the participant was considered "not retained in the study" for that time point. The survival

**Table 2. Baseline characteristics of enrolled, retained, and virally suppressed HIV positive TGW.**

| Characteristic | Enrolled patients (N = 61) | Retained at week 48 (n = 47) | Virally suppressed at week 48 (n = 44) |
|---|---|---|---|
| Age, median (IQR) | 28 (25–32) | 28 (26–33) | 28 (26–33) |
| Foreign-born | 18 (30%) | 14 (30%) | 13 (30%) |
| Incomplete high school education or less | 37 (61%) | 27 (57%) | 25 (57%) |
| Unstable housing | 32 (53%) | 28 (60%) | 26 (59%) |
| Sex work (current) | 47 (77%) | 35 (75%) | 32 (73%) |
| pVL HIV, median copies/ml (IQR) | 46,908 (12,627–275,316) | 58,161 (13,230–308,994) | 78,675 (12,928–300,282) |
| CD4 count, median (IQR) | 383 (230–601) | 389 (216–614) | 391 (235–615) |
| pVL HIV >100,000 c/mL, n(%) | 25 (41%) | 22 (47%) | 21 (48%) |
| CD4 count <200, n (%) | 12 (20%) | 9 (19%) | 8 (18%) |
| Current syphilis | 22 (36%) | 19 (40%) | 17 (39%) |
| Hepatitis B | 1 (2%) | 1 (2%) | 1 (2%) |
| Hepatitis C | 3 (5%) | 2 (4%) | 2 (5%) |
| Use of gender-affirming hormone therapy | 8 (13%) | 8 (17%) | 8 (18%) |
| Suicidal ideation | 16 (26%) | 13 (28%) | 13 (30%) |
| Obesity/overweight | 26 (44%) | 19 (42%) | 18 (43%) |
| Hazardous alcohol use | 32 (53%) | 24 (51%) | 22 (50%) |
| Drug abuse | 8 (13%) | 6 (13%) | 5 (11%) |
| Cocaine use (last year) | 32 (53%) | 24 (51%) | 22 (50%) |

Abbreviations: IQR, interquartile; pVL, plasma viral load

analysis performed shows that the probability of retention in the study decreases with time (Fig 2). In the first month after enrollment (week 4), the mean retention probability is 92%. By month 2 (week 8) it descends to 87%; in month 3 (week 12) it descends to 81%; and in month 6 (week 24) it is 73%. The mean retention probability by month 9 (week 36) is 68% and by the end of the study (week 48) it is 65%. The greater decrease in retention probability occurred in the first to third month of treatment.

## Safety and tolerability

The study drugs were well tolerated. Nine clinical AEs (grade 1: 17% [8/47] and grade 2: 2% [1/47]) were classified as possibly related to study treatment. The most frequent laboratory abnormalities were elevated aspartate transaminase/alanine transaminase (grade 2: 20% [9/45]) without an increase in bilirubin. Hypertriglyceridemia (grade 2) was observed in 13.3% (6/45) of the participants.

Five participants presented SAEs, although none were considered related to study treatment, and two resulted in the participant's discontinuation from the study (one was a suicide attempt after 8 weeks of treatment and the other, a meningeal tuberculosis at week 4, leading to death). Although external factors were identified in the case of suicidal attempt, after discussing with the medical team, it was decided to discontinue ART. The other 3 SAEs included cryptococcal meningitis at day 5, intestinal tuberculosis at week 20, and appendicitis at week 24 that resulted in hospital admission. These three participants finalized the protocol.

Given that weight gain has been reported for DTG-based regimens, changes at week 48 were analyzed for exploratory reasons. Participants increased their weight, on average, by 2.6% compared to their weight at the baseline visit. In absolute terms, there was an increase of 1.5 kilos on average between baseline and week 48; this difference was not statistically significant ($z = -1.934$, $p = 0.053$), although it can be considered a trend ($p$ value $< 0.10$).

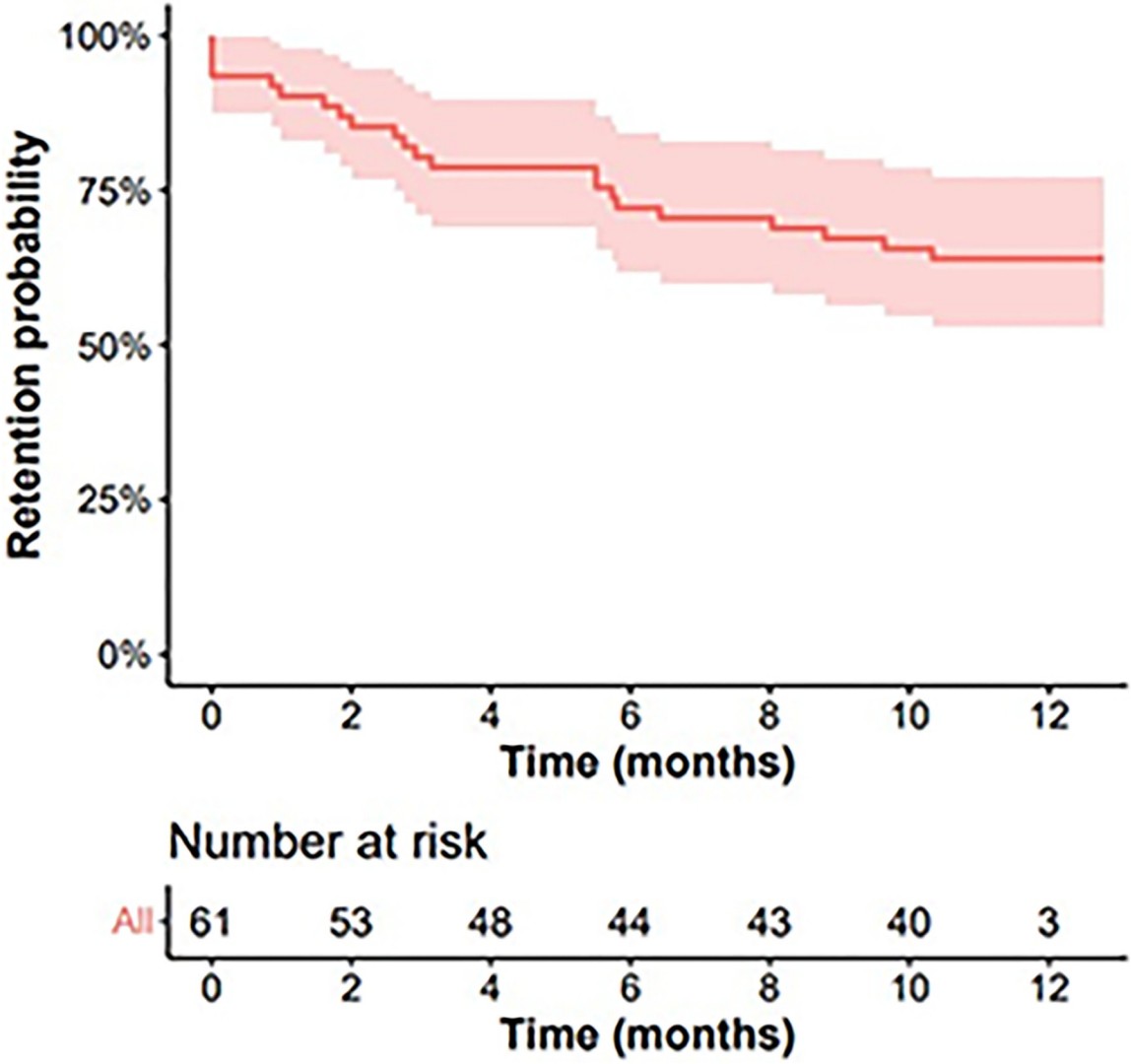

**Fig 2. Time to last visit/medication dispensed curve.**

### Clinical and psychosocial factors associated with retention

As shown in Table 3, no statistically significant differences were observed for most baseline clinical and psychosocial variables between TGW retained and not retained in the study at week 48. Differences in age were significant *(p = 0.016)*, participants retained in the study are older than not retained participants. Unstable housing was also significant *(p = 0.041)*, participants retained in the study report unstable housing in a greater proportion than not retained participants.

## Discussion

This study with HIV-1-infected ART-naïve TGW showed that 77% met the protocol definition of retention (at week 48) after their initial prescription of drug regimen comprising DTG plus TDF-FTC or 3TC. Likewise, 72% and 97% of participants showed viral suppression at week 48 in ITT and per-protocol analyses, respectively. The regimen had low toxicity, good tolerance and high adherence (about 95%).

**Table 3. Baseline clinical and psychosocial characteristics by retention status among HIV positive TGW.**

| | Retained (n = 47) | Not Retained (n = 14) | Mann-Whitney | $x^2$ | $p$ |
|---|---|---|---|---|---|
| Age, median (IQR) | 28 (26–33) | 24 (23–29) | 189.5 | - | 0.016* |
| Foreign-born | 14 (30%) | 4 (29%) | - | ** | 1.000 |
| Incomplete high school education or less | 27 (57%) | 10 (71%) | - | 0.884 | 0.347 |
| Unstable housing | 28 (60%) | 4 (29%) | - | 3.395 | 0.041* |
| Sex work (current) | 35 (75%) | 12 (86%) | - | ** | 0.488 |
| pVL HIV, median copies/ml (IQR) | 58,161 (13,230–308,994) | 26.031 (9,610–71,135) | 258.0 | - | 0.223 |
| CD4 count, median (min/max) (IQR) | 389 (20–1118) (216–614) | 374 (82–116) (257–528) | 318.0 | - | 0.850 |
| pVL HIV >100,000 c/mL, n(%) | 22 (47%) | 3 (21%) | - | 2.873 | 0.090 |
| CD4 count <200, n (%) | 9 (19%) | 3 (21%) | - | ** | 1.000 |
| Current syphilis | 19 (40%) | 3 (21%) | - | 1.688 | 0.194 |
| Hepatitis B | 1 (2%) | 0 (0%) | - | ** | 1.000 |
| Hepatitis C | 2 (4%) | 1 (7%) | - | ** | 0.549 |
| Use of gender-affirming hormone therapy | 8 (17%) | 0 (0%) | - | ** | 0.180 |
| Suicidal ideation | 13 (28%) | 3 (21%) | - | ** | 0.742 |
| Obesity/overweight | 19 (42%) | 7 (50%) | - | 0.262 | 0.609 |
| Hazardous alcohol use | 24 (51%) | 8 (57%) | - | 0.160 | 0.689 |
| Drug abuse | 6 (13%) | 2 (14%) | - | ** | 1.00 |
| Cocaine use (last year) | 24 (51%) | 8 (57%) | - | 0.160 | 0.689 |

* $p < 0.05$

** Fisher's exact test is recorded.

Although retention and viral suppression figures are below the UNAIDS recommended goals, they still represent an achievement in this challenging group of patients. In fact, studies from Latin America show lower rates of retention among TGW. A respondent-driven sampling study among TGW in Brazil that aimed to estimate the HIV cascade of continuum of care showed that 67.2% reported linkage to care [26]. In Peru, retention among men who have sex with men and TGW diagnosed with HIV was estimated in 65% [27]. Moreover, the results of the current study showed better retention than the estimate found in a retrospective study conducted by our research group in a public hospital in Buenos Aires, reporting a retention of 46.4% at 1 year of treatment initiation [28]. Likewise, the proportion found in this study was also higher than the one found in the general population of Buenos Aires (65.5%) [29]. It is worth noting that when these previous studies were conducted, ART combinations did not include DTG as the first-line, preferred treatment. It is possible that the study regimen exhibited a better tolerability than the standard first-line at the time of designing this protocol. Additionally, unlike these previous studies, in this trial, treatment was provided in the context of a trans-affirmative care service. This characteristic may have contributed to increased retention in the study.

Baseline characteristics of the sample show a high level of psychosocial vulnerability. One third of the participants (30%) were foreign-born, half of them reported unstable housing, and a high proportion was engaged in sex work for a living. These factors, among others, may increase mobility and, therefore, the risk of being lost to follow-up in HIV care [30]. Simplification of ART by preferring once-daily uptake regimens, such as the one including DTG, may contribute to compensate for this situation and to explain the proportion of participants retained in this study.

Age was one of the factors associated with retention in this sample of TGW. Older TGW are more likely to be retained in the study than younger ones. In relation to this, older TGW

may have a history of approaching the healthcare system seeking gender-affirming medical procedures (e.g., hormone therapy), which then facilitates access to other healthcare services that they may need, such as HIV care [31, 32]. Participants in this study did not report high proportions of use of hormone therapy at baseline. Nonetheless, those who were retained were the ones who did report use of hormones for gender-affirmation, which constitutes the first contact or the entrance to the healthcare system. Additionally, older TGW could have more confidence in the healthcare team and a better and more trustworthy patient-provider relationship, which contribute to improved retention [33]. Even though substance use has been identified as a factor that negatively affects retention among TGW [34], in our study, the use of alcohol and cocaine in the last year was not significantly associated with retention. However, the high levels of substance use in this population, as shown in these results, is another factor to take into consideration when deciding to prescribe an ART regimen with less potential for interaction.

Adherence to ART is affected by a range of factors, among them, the AEs associated with chronic therapy [35]. This study supports the current WHO recommendations regarding the use of DTG-based therapies and, in line with other studies [36–39] this regimen showed excellent safety and good tolerance with adherence percentages of 95%. Five patients presented SAEs, though they were unrelated to the study treatment and, on the contrary, the majority corresponded to opportunistic infections that reflected the low immunological level due to advanced HIV infection. Some of these infections, especially those associated with tuberculosis, may have emerged in the context of a possible immune reconstitution inflammatory syndrome, due to initiation of ART with a potent combination of drugs, in a predisposing immunologic profile.

The absence of a control arm did not allow us to draw conclusions about efficacy of once-daily DTG-based ARV-naïve patients. However, in this single-arm study, this regimen achieved viral suppression in 72% of 45 patients by ITT analysis and 97% in the analysis per-protocol. Given that the number of "non-suppressed" participants was very small, an exploratory analysis of possible correlates of this condition could not be conducted, as required statistical parameters were not achieved.

In the course of DTG-based ART, an unexpected excess in weight gain has been reported [15, 40]. Although results showed an increase between the baseline visit and week 48 of treatment, it was non-significant, and less than that reported in other studies.

In Argentina, gender identity stigma from healthcare workers is associated with avoidance of TGW to attend healthcare services [7]. Our study was conducted by a trans-competent team in a trans-affirmative HIV care service, which, in addition to the proposed ART, may have contributed to the high retention rates among participants. The presence of transgender peer navigators may have also contributed to a better retention. As previous local research suggested, a trans-affirmative healthcare service can have not only a medical benefit, but also a psychosocial gender-affirming effect that positively impacts general health and wellbeing [11]. Future research should compare retention in different healthcare settings, considering the mediating role of a trans-affirmative approach in HIV care and the inclusion of peer navigators as part of the healthcare team.

## Limitations

The present study has some limitations. Firstly, our findings cannot be generalized to all HIV positive TGW in Argentina. A small non-probability sample was enrolled, limiting generalizability of results. Still, the final sample showed sociodemographic characteristics that are similar to nationwide studies with non-clinical samples [41]. Secondly, ART adherence was

measured through self-report and therefore, this measure may not be free of social desirability bias. Lack of drug dosage and lack of uniformity with other studies limited comparability of results across contexts.

Although our results seem favorable in the context of this specific population, they are still far from the achievement of the UNAIDS global 95-95-95 HIV goals by 2030 [42], especially in a population with such high exposure to HIV and engagement in sex work.

## Conclusion

In this study, DTG-based treatment delivered by a trans-competent team in a trans-affirmative service was safe and well tolerated by TGW and associated with high retention, high adherence and high viral suppression at 48 weeks among those being retained. Thus, results support the use of DTG-based treatments in this population, as it is recommended by the WHO [14] in naïve patients in general.

Despite involving a small sample of TGW, this is the first longitudinal study of these characteristics in Argentina, contributing to the scarce information available in Latin America, on retention and adherence to ART among transgender people. It also supports the need for a more comprehensive approach in HIV care, integrating medical and psychosocial/behavioral factors, to address retention in this population. As these results suggest, a trans-affirmative approach in care and the inclusion of peer navigators may facilitate access to healthcare and favor greater retention. Thus, it is recommended to continue exploring the feasibility of the implementation of these strategies in public healthcare services.

## Supporting information

**S1 File. Original protocol Spanish version.**
(PDF)

**S2 File. Original protocol English version.**
(PDF)

**S3 File. Minimal anonymized data set.**
(XLSX)

## Acknowledgments

The authors would like to thank all the participants and organizations that work with us, and that collaborate in improving the health of the trans population: Asociación de Travestis, Transexuales y Transgéneros de Argentina (A.T.T.T.A), Asociación Civil Hotel Gondolin and Casa Trans.

## Author Contributions

**Conceptualization:** Claudia E. Frola, Inés Aristegui, María I. Figueroa, Nadir Cardozo, Virginia Zalazar, Pedro Cahn, Omar Sued.

**Data curation:** Claudia E. Frola, Inés Aristegui, María I. Figueroa, Pablo D. Radusky, Virginia Zalazar, Carina Cesar, Valeria Fink, Pedro Cahn, Omar Sued.

**Formal analysis:** Claudia E. Frola, Inés Aristegui, María I. Figueroa, Pablo D. Radusky, Virginia Zalazar, Carina Cesar, Valeria Fink, Pedro Cahn, Omar Sued.

**Funding acquisition:** Pedro Cahn, Omar Sued.

**Investigation:** Claudia E. Frola, Inés Aristegui, María I. Figueroa, Pablo D. Radusky, Nadir Cardozo, Virginia Zalazar, Carina Cesar, Patricia Patterson, Valeria Fink, Ana Gun, Pedro Cahn, Omar Sued.

**Methodology:** Claudia E. Frola, Inés Aristegui, María I. Figueroa, Pablo D. Radusky, Nadir Cardozo, Virginia Zalazar, Carina Cesar, Patricia Patterson, Valeria Fink, Ana Gun, Pedro Cahn, Omar Sued.

**Project administration:** Claudia E. Frola, Omar Sued.

**Resources:** Claudia E. Frola, Inés Aristegui, María I. Figueroa, Pablo D. Radusky, Ana Gun, Pedro Cahn, Omar Sued.

**Supervision:** Claudia E. Frola, María I. Figueroa, Nadir Cardozo, Patricia Patterson, Pedro Cahn, Omar Sued.

**Validation:** Claudia E. Frola, Inés Aristegui, María I. Figueroa, Pedro Cahn, Omar Sued.

**Writing – original draft:** Claudia E. Frola, Inés Aristegui, María I. Figueroa, Pablo D. Radusky, Nadir Cardozo, Virginia Zalazar, Carina Cesar, Patricia Patterson, Valeria Fink, Ana Gun, Pedro Cahn, Omar Sued.

**Writing – review & editing:** Claudia E. Frola, Inés Aristegui, María I. Figueroa, Pablo D. Radusky, Pedro Cahn, Omar Sued.

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
