## [Decision Letter · Decision Letter 0]

12 Oct 2022

PONE-D-22-15918Retention in care among transgender women treated with Dolutegravir associated with Tenofovir/Lamivudine or Emtricitabine in Argentina: TransViiV studyPLOS ONE

Dear Dr. Frola,

Thank you for submitting your manuscript to PLOS ONE. After careful consideration, we feel that it has merit but does not fully meet PLOS ONE’s publication criteria as it currently stands. Therefore, we invite you to submit a revised version of the manuscript that addresses the points raised during the review process.

We look forward to receiving your revised manuscript.

Kind regards,

Patricia Evelyn Fast, MD, Ph.D.

Academic Editor

PLOS ONE

Journal Requirements:

2. Thank you for submitting your clinical trial to PLOS ONE and for providing the name of the registry and the registration number. The information in the registry entry suggests that your trial was registered after patient recruitment began. PLOS ONE strongly encourages authors to register all trials before recruiting the first participant in a study. As per the journal’s editorial policy, please include in the Methods section of your paper: 1) your reasons for your delay in registering this study (after enrolment of participants started); 2) confirmation that all related trials are registered by stating: “The authors confirm that all ongoing and related trials for this drug/intervention are registered”.

3. Please mention the trial registration number of your clinical trial in the abstract of your manuscript.

5. We note that the original protocol file you uploaded contains a confidentiality notice indicating that the protocol may not be shared publicly or be published. Please note, however, that the PLOS Editorial Policy requires that the original protocol be published alongside your manuscript in the event of acceptance. Please note that should your paper be accepted, all content including the protocol will be published under the Creative Commons Attribution (CC BY) 4.0 license, which means that it will be freely available online, and any third party is permitted to access, download, copy, distribute, and use these materials in any way, even commercially, with proper attribution.

Therefore, we ask that you please seek permission from the study sponsor or body imposing the restriction on sharing this document to publish this protocol under CC BY 4.0 if your work is accepted. We kindly ask that you upload a formal statement signed by an institutional representative clarifying whether you will be able to comply with this policy. Additionally, please upload a clean copy of the protocol with the confidentiality notice (and any copyrighted institutional logos or signatures) removed.

Additional Editor Comments:

Please carefully read the three reviews provided. Each of the reviewers has identified ways in which the presentation or interpretation can be made much clearer.  Although additional analyses are suggested, the criticism that must be addressed is that the paper is not always clear on what was done (these are generally minor comments), and that the interpretation may not always be supported by the data.  In particular, please clarify the difference between retention in care and retention in the study.

 I look forward to your response.

Reviewers' comments:

Reviewer's Responses to Questions

**Comments to the Author**

1. Is the manuscript technically sound, and do the data support the conclusions?

Reviewer #1: Partly

Reviewer #2: Yes

Reviewer #3: Yes

2. Has the statistical analysis been performed appropriately and rigorously? 

Reviewer #1: Yes

Reviewer #2: Yes

Reviewer #3: Yes

3. Have the authors made all data underlying the findings in their manuscript fully available?

Reviewer #1: Yes

Reviewer #2: Yes

Reviewer #3: Yes

4. Is the manuscript presented in an intelligible fashion and written in standard English?

Reviewer #1: Yes

Reviewer #2: Yes

Reviewer #3: Yes

5. Review Comments to the Author

Reviewer #1: Thank you for the opportunity to review this paper. The authors present findings from a study of the HIV care and treatment continuum among transgender women in Buenos Aires, Argentina. While the overall purpose of the study appears to be to conduct a trial of a new treatment, they emphasize the significance of the longitudinal findings given that there is limited information about HIV care and treatment outcomes over time among trans women.

The findings of HIV care and treatment outcomes are indeed a great contribution of the paper; however, the limited information on the overall context of the study, services, and measures diminishes the interpretation and impact of the findings. There is some tension/confusion for me as to whether this is really being presenting as the results of a trial of the new treatment approach or as an observational study of trans women living with HIV in a trans-competent clinic. I suggest being more explicit about this and providing more context about the services and care that was offered beyond the new drug regimen. It would also be helpful to provide more commentary on how the landscape of HIV treatment has changed since this trial was completed in 2018, especially for non-clinical readers, and how these findings fit into current treatment practice.

Abstract:

Rather than focusing on just the disproportionate burden of HIV among trans women, given the focus of this paper on HIV care and treatment outcomes, it would be more relevant to discuss the gaps in retention, adherence, and viral suppression that trans women experience.

Suggest including the years of the study in the abstract.

Intro

I would strongly suggest editing the first paragraph to provide more structure. The first paragraph is nearly the entire first page and covers a huge range of information. Consider using shorter pargraphs to address the key points around: 1) the overall burden of HIV; 2) the importance of TASP and gaps among trans women; and 3) psychosocial and contextual factors to consider.

It is not clear why there is discussion of migrants in the first paragraph. Suggest focusing on the target population of trans women with HIV. The role of mobility can be picked up the discussion but it feels out of place in the intro.

Methods

In the study setting section, there is insufficient detail on what the trans-competent care entails and how it is delivered as well as the peer navigation. While the study is framed as a drug trial, these are interventions that were delivered, and they are being interpreted in the findings. We don’t really get a clear idea of whether it is the new drug regimen, the peer support, or the quality of care that is most important (or all of them together?) because there is very minimal discussion of how the care was provided in the methods section.

How did trans women self-identify as part of the recruitment and enrollment process? More detail on this would be helpful.

As worded (“treatment was provided on site”), it sounds like participants got daily treatment on site – every day. Suggest rewording to reflect the intervals between their study visits and other clinical care visits.

Overall, as described, it is not clear that HIV care and treatment variables were measured at 24 and 48 weeks since some of the definitions only refer to 48 weeks. Suggest reviewing descriptions of measures for clarity and consistency with the presentation of results.

Related to this, retention was defined as being “retained in care through 48 weeks”. This is really a measure of study retention, meaning participants came for the that study visit, but it does not fully reflect retention in HIV care, which is how it is being interpreted in the discussion. Additionally, this outcome is also reported for 24 weeks. It would be more informative to report on retention in HIV care based on a more standard def of attendance at clinic visits. If this information was not collected, the interpretation should be modified to be consistent with what was measured.

Were participants provided compensation? This is not included in the ethics section. It could be influential to retention and should be include as part of the ethical section of the paper.

Results

Results on suicidal ideation are in the table but not the text. What support was provided for participants who were found to have suicidal ideation?

Given small sample size, it would be helpful to consistently present the ns as well as % in the presentation of results.

For those lost to follow-up, it is not clear when their last visit was. How frequently were patients coming in?

Acronyms need to be spelled out (ex, PDVF, SAE). This is broadly focused journal with clinical and non-clinical readers, and these are not universally known acronyms, which limits the accessibility of the paper.

Discussion

As noted above, results are interpretated treating study retention as the same as HIV care retention. This needs to be addressed.

It would be helpful to situate these findings more explicitly in the context of current treatment practice.

Given the design, I do not consider the characteristics of the site to be a limitation. It seems that the argument of the paper is that integrated approaches are needed and this study is providing evidence to support that. Therefore, I would not call the site a limitation.

In order to fully appreciate the discussion of an integrated approach in the discussion, as noted above, we need to know more about what services were provided and how they were delivered.

Reviewer #2: Authors describe a pilot study was to determine retention in care, adherence and viral suppression in naïve Transgender women (TGW) starting a DTG-based first line antiretroviral treatment (ART) and to identify clinical and psychosocial factors associated with retention.

They designed a prospective, open-label, single-arm trial among ART-naïve HIV positive TGW. Participants were followed at weeks 4, 8, 12, 24, 36 and 48, in a trans-sensitive HIV service that included peer navigators. Retention was defined as the proportion of TGW retained at week-48 and adherence was self-reported.

Of 75 TGW screened, 61 were enrolled. At week 48, 77% were retained and 72% had viral suppression. Older age was associated with better retention. DTG-based treatment delivered by a trans-competent team in a trans-affirmative service was safe and well tolerated by TGW and associated with high retention.

While the WHO has recommended ART regimens for treatment naïve people with HIV to be Dolutegravir-based.

As a possible disadvantage, only an increased risk of weight gain in the short term has

been reported in those starting DTG-based regimens.

Authors describe a 48-week pilot study – and aim to identify clinical and psychosocial factors

associated with retention in care.

Comments:

This is a well written observational study of ART-naïve TGW who are initiated on ART by an experienced clinical trial team – sensitive to trans-challenges.

I have really very little to remark.

Why do authors call this a pilot study? Will they propose a large efficacy trial? I didn’t see the arguments for this in the discussion.

In lines 280-286 – authors present data on retention from other studies and show that retention in the present study is above that of other studies reported. I would have expected nothing less from an experienced trial team – supported by Trans peer mobilisors.

In fact, in the title of the manuscript – “retention in care” is emphasized. I wonder if retention is the main study outcome. Isn’t the study a description of care cascade indicators?

I suggest a title change, replacing “retention in care” with “Care cascade indicators” in…..

End.

Reviewer #3: General comments:

The analyses of this trial are pretty straightforward and I don't see any changes needed. Though, given the longitudinal data collection performed on this participants, it is disappointing that the authors did not perform and present longitudinal analyses of this cohort. Potentially a survival analysis would be interesting to determine how long participants were retained.

The more problematic parts of this study are related to the study design. The authors mention in lines 326-327 that this single-arm trial will not allow for conclusions about efficacy and that the study lacks generalizability (lines 344-345). Though it struck me that the first half of the discussion (approximately lines 273-299) were about the efficacy of the trial and comparing the results to other results. While I think it's important to compare the results with other studies, it behooves to provide some context as to why these are different. The retention and viral suppression results could be due to the intervention, but could also be due to differences in the population. For instance, take this statement (lines 284-285):

Moreover, the results of the current study showed better retention than the estimate found in a retrospective study conducted by our research group in a public hospital of Buenos Aires, reporting a retention of 46.4% at 1 year of treatment initiation [25].

As the authors note, the differences could be due to differences in the first line treatment, but could also be due to differences in the population. The authors should provide more information on where these results are coming from and why they might be different. For instance, the authors found an association with age in their data; what were the age distributions in other studies? Maybe worse outcomes would be expected in those studies given the age distributions.

Finally, given this is a pilot or proof of concept study, I think the authors should provide more information on how these results should be used. Are further studies warranted? If so, how should they be designed? Who should be the target population?

Specific comments:

1. (line 174) I am not sure what "independent samples" means in this statement.

2. (line 300) "The more significantly characteristic" does not make sense.

6. PLOS authors have the option to publish the peer review history of their article (what does this mean?). If published, this will include your full peer review and any attached files.

Reviewer #1: No

Reviewer #2: No

Reviewer #3: No

---

## [Author Response · Author response to Decision Letter 0]

25 Nov 2022

Patricia Evelyn Fast, MD, Ph.D.

Academic Editor

PLOS ONE

We thank the Editorial Office of the PLOS ONE and the reviewers for their helpful comments. We feel that the manuscript has improved as a result of the comments we received.

Below we have outlined the changes made to the manuscript in response to the reviewers’ comments. Each response follows the comment it is meant to address, and any new text has been noted in bold font. All changes in the manuscript have been marked in the text, with track changes.

Journal Requirements

Response: We have thoroughly revised the manuscript so that it meets the journal’s style requirements.

2. Thank you for submitting your clinical trial to PLOS ONE and for providing the name of the registry and the registration number. The information in the registry entry suggests that your trial was registered after patient recruitment began. PLOS ONE strongly encourages authors to register all trials before recruiting the first participant in a study. As per the journal’s editorial policy, please include in the Methods section of your paper: 1) your reasons for your delay in registering this study (after enrolment of participants started); 2) confirmation that all related trials are registered by stating: “The authors confirm that all ongoing and related trials for this drug/intervention are registered”.

Response: Thank you for noting that our enrollment started before the registry in www.clinicaltrials.org. Please note that in Argentina, local regulations require to register trials in the national registry ReNIS (Registro Nacional de Investigaciones en Salud, https://www.argentina.gob.ar/salud/registroinvestigaciones) if:

a) the study is carried out with federal funding

b) the study is a phase I, II or III,

c) the study sought a new indication, or dose registration

d) the study involves bioequivalence

Although registration of this study was not required, we decided to register it as, at that time, we were aware that registration would become a requirement for publishing. Currently, all the studies with drugs or interventions are being registered.

As requested, we have included a paragraph in the Study design section (paragraph 3), clarifying the reasons for the delay in registering this study after enrollment of participants started, and incorporating a statement, confirming that all related trials are registered. This paragraph is reproduced below:

“According to local regulations, the registration of this study in the national registry of clinical trials (ReNIS - Registro Nacional de Investigaciones en Salud) was not required. However, we decided to register it as we were aware that it was a requirement for the publication of the study’s results. Consequently, it was registered after the enrollment of participants started, under Clinical Trial Number NCT03033836. The authors confirm that all ongoing and related trials for this drug/intervention are registered.”

3. Please mention the trial registration number of your clinical trial in the abstract of your manuscript. 

Response: We have mentioned the registration number of our clinical trial in the Abstract (line 9).

Response: We apologize for our mistake when choosing the option “data set available upon request). There are no restrictions on sharing our de-identified data set. We uploaded the anonymized data set with this revision, as supporting information. 

5. We note that the original protocol file you uploaded contains a confidentiality notice indicating that the protocol may not be shared publicly or be published. Please note, however, that the PLOS Editorial Policy requires that the original protocol be published alongside your manuscript in the event of acceptance. Please note that should your paper be accepted, all content including the protocol will be published under the Creative Commons Attribution (CC BY) 4.0 license, which means that it will be freely available online, and any third party is permitted to access, download, copy, distribute, and use these materials in any way, even commercially, with proper attribution.

Therefore, we ask that you please seek permission from the study sponsor or body imposing the restriction on sharing this document to publish this protocol under CC BY 4.0 if your work is accepted. We kindly ask that you upload a formal statement signed by an institutional representative clarifying whether you will be able to comply with this policy. Additionally, please upload a clean copy of the protocol with the confidentiality notice (and any copyrighted institutional logos or signatures) removed. 

Response: We apologize for this, as well, as the confidentiality notice no longer applies and should have been removed before submission. There are no restrictions for publication of the protocol. We have removed the confidentiality notice and all institutional logos and signatures.

Editor Comments

Please carefully read the three reviews provided. Each of the reviewers has identified ways in which the presentation or interpretation can be made much clearer. Although additional analyses are suggested, the criticism that must be addressed is that the paper is not always clear on what was done (these are generally minor comments), and that the interpretation may not always be supported by the data. In particular, please clarify the difference between retention in care and retention in the study.

Response: We have carefully gone through each of the reviewers’ comments, in order to address them. We believe that these comments have greatly contributed to this manuscript’s improvement. Below you will find a detailed explanation on how each comment was addressed and responded.

Reviewer #1 

The findings of HIV care and treatment outcomes are indeed a great contribution of the paper; however, the limited information on the overall context of the study, services, and measures diminishes the interpretation and impact of the findings. There is some tension/confusion for me as to whether this is really being presenting as the results of a trial of the new treatment approach or as an observational study of trans women living with HIV in a trans-competent clinic. I suggest being more explicit about this and providing more context about the services and care that was offered beyond the new drug regimen. It would also be helpful to provide more commentary on how the landscape of HIV treatment has changed since this trial was completed in 2018, especially for non-clinical readers, and how these findings fit into current treatment practice.

Response: We thank the reviewer for the valuable feedback and comments. We agree that the manuscript would benefit from greater explanation of the context of the study. We have explained the context in 2015, regarding HIV treatment, that led to the design of this study in the Study design section, paragraph 1, lines 2-5, including the following fragment:

“This study was designed in a national context of use of efavirenz-containing triple antiretroviral regimen with possible changes in sleep quality. This regimen could particularly affect a population with high proportions of engagement in nightly sex work, such as TGW, therefore negatively impacting adherence and retention.”

In the Introduction section, we describe the current practice in HIV treatment, in paragraph 2, in the following fragment:

“Transition to integrase inhibitors (INIs) based regimens is associated with lower discontinuation rate and higher efficacy to increase viral suppression and, potentially, higher retention in HIV care [18]. For this reason, currently the World Health Organization (WHO) [19] recommends dolutegravir (DTG) with two NRTIs as the preferred ART for naïve individuals based on its better efficacy and safety profile, in order to reduce adverse events, improve adherence and retention in HIV care and as a response to increments of the primary resistance to non-nucleoside reverse transcriptase inhibitors.” 

In the Conclusion section, paragraph 1, lines 4-5, we suggest how these findings fit into current treatment practice:

“Thus, results support the use of DTG-based treatments in this population, as it is recommended by the WHO [19] in naïve patients in general.”

The rest of the points will be responded in the following in the response to the following comments.

Abstract: Rather than focusing on just the disproportionate burden of HIV among trans women, given the focus of this paper on HIV care and treatment outcomes, it would be more relevant to discuss the gaps in retention, adherence, and viral suppression that trans women experience.

Response: We agree with the reviewer, so we have modified the first two sentences of the abstract to emphasize the gaps in retention, adherence and viral suppression, as follows:

“In Argentina, transgender women (TGW) have a high HIV prevalence (34%). However, this population shows lower levels of adherence, retention in HIV care and viral suppression than cisgender patients.”

Suggest including the years of the study in the abstract.

Response: We have included the years of the study in the abstract, lines 10-11.

Introduction: I would strongly suggest editing the first paragraph to provide more structure. The first paragraph is nearly the entire first page and covers a huge range of information. Consider using shorter paragraphs to address the key points around: 1) the overall burden of HIV; 2) the importance of TASP and gaps among trans women; and 3) psychosocial and contextual factors to consider.

Response: We have reorganized the whole Introduction section with shorter paragraphs and a clearer order of the information, for greater focus and clarity. We have organized the section with the following order: burden of HIV among TGW and gaps in retention and adherence to ART (paragraph 1), treatment-related factors that impact retention and adherence and the benefit of a dolutegravir-based regimen (paragraph 2), importance of increasing retention and adherence among TGW (including TasP) (paragraph 3) and aims and hypotheses (paragraph 4). We aimed to maintain the Introduction section as concise as possible, while providing it with more focus. 

It is not clear why there is discussion of migrants in the first paragraph. Suggest focusing on the target population of trans women with HIV. The role of mobility can be picked up the discussion but it feels out of place in the intro.

Response: We agree with the reviewer. Thus, we have removed the reference to mobility and migrants from the Introduction.

Methods: In the study setting section, there is insufficient detail on what the trans-competent care entails and how it is delivered as well as the peer navigation. While the study is framed as a drug trial, these are interventions that were delivered, and they are being interpreted in the findings. We don’t really get a clear idea of whether it is the new drug regimen, the peer support, or the quality of care that is most important (or all of them together?) because there is very minimal discussion of how the care was provided in the methods section.

Response: We agree with the reviewer’s comment. We have included a detailed description of what our trans-affirmative service entails and what the main role of the peer navigators is, including their primary tasks. We have listed the main components that make our healthcare service trans-affirmative. The fragment containing this information was included in the Study setting section, paragraph 1, and it is reproduced below:

“Our trans-affirmative healthcare service includes a) use of patients’ preferred name and pronoun in interactions, clinical records and forms (which include sex assigned at birth and gender identity); b) an interdisciplinary trans-competent trained staff, aware of transgender people’s needs and accepting of their identities; c) integration of multiple services for this community (e.g., HIV, gender-affirming medical procedures, anal health) to simplify service delivery; d) adjustment to transgender populations’ social contexts (e.g., flexible scheduling and hours); and e) inclusion of transgender peer navigators. Peer navigators function as a bridge between the research site and the transgender community. Some of their main tasks are: a) to provide health information to their peers adapting it to their community and making it more accessible and comprehensible, b) to invite potential participants and to enroll them in the studies, c) to verify that they understood the informed consent correctly and to answer any concern about it, d) to assist transgender people in obtaining medical appointments and in navigating the healthcare service, e) to remind participants their upcoming visits, and f) to contact lost to follow-up participants to re-engage them in the study.”

How did trans women self-identify as part of the recruitment and enrollment process? More detail on this would be helpful.

Response: We understand that the reviewer refers to how our team knew that participants were TGW. We have included greater specification and detail in the description of the recruitment and enrollment procedures, in the Study design section, paragraph 2. We highlight below the fragments that were added:

“Participants were recruited by outreach efforts of peer navigators, through testing campaigns conducted in places were transgender people gather or live, and through collaboration with a local transgender community-based organization.”

Primarily, participants were recruited in activities exclusively oriented to the transgender community or in places that are mostly exclusive for this population. Also, peer navigators were able to identify their peers and offer them to be a participant of the study.

As worded (“treatment was provided on site”), it sounds like participants got daily treatment on site – every day. Suggest rewording to reflect the intervals between their study visits and other clinical care visits.

Response: For greater clarity, we have reworded that fragment to explicitly detail the visits that the study entailed, in the following way:

“Treatment was provided on site at the following visits: baseline and weeks 4, 8, 12, 24, 36 y 48.” 

Overall, as described, it is not clear that HIV care and treatment variables were measured at 24 and 48 weeks since some of the definitions only refer to 48 weeks. Suggest reviewing descriptions of measures for clarity and consistency with the presentation of results.

Response: In the Measures section, we have revised that the description of each measure for the HIV care and treatment variables (retention, adherence and viral suppression), it was mentioned at which visit it was conducted (retention at weeks 24 and 48, adherence at each study visit –including weeks 24 and 48-, and viral suppression at week 48). This information was added where there was no mention or where it was incomplete. We have checked consistency between this information and the presentation of results.

Related to this, retention was defined as being “retained in care through 48 weeks”. This is really a measure of study retention, meaning participants came for the that study visit, but it does not fully reflect retention in HIV care, which is how it is being interpreted in the discussion. Additionally, this outcome is also reported for 24 weeks. It would be more informative to report on retention in HIV care based on a more standard def of attendance at clinic visits. If this information was not collected, the interpretation should be modified to be consistent with what was measured.

Response: We agree with the reviewer regarding this point. We have actually measured “retention in the study” and not “retention in HIV care”. Information on “retention in HIV care” was not collected, unfortunately. We have corrected the definition of the variable in the Measures section. We have also thoroughly revised the entire manuscript to remove any ambiguity regarding what was actually measured. We have revised the interpretation of this result in the Discussion section so that it is consistent with what was measured.

Were participants provided compensation? This is not included in the ethics section. It could be influential to retention and should be include as part of the ethical section of the paper.

Response: We have included a fragment clarifying this point in the Ethics statement section, as follows:

“Participation was voluntary. At each visit participants received a $150 Argentine pesos compensation (approximately, 15 USD at the moment of the study) to cover transportation costs, and coupon exchangeable for a basic breakfast or meal.”

Results: Results on suicidal ideation are in the table but not the text. What support was provided for participants who were found to have suicidal ideation?

Response: We have included results on suicidal ideation also in the text, in the Results section, paragraph 3, lines 3-4, in the following fragment:

“Regarding mental health indicators, 26% (n=16) of these TGW reported significant suicidal ideation in the last 2 weeks.”

Additionally, in the Study design section, paragraph 2, lines 5-7, we included a fragment describing the procedures that were followed with participants who reported significant suicidal ideation. It is reproduced below:

“Participants with significant suicidal ideation were assessed for suicide risk by a mental healthcare provider, and referral to mental health services was facilitated to those who required it.”

Given small sample size, it would be helpful to consistently present the ns as well as % in the presentation of results.

Response: We thank the reviewer for noting this. In the report of the baseline clinical and psychosocial characteristics, in the Results section, we have reported the n in text, next to the corresponding proportion.

For those lost to follow-up, it is not clear when their last visit was. How frequently were patients coming in?

Response: As mentioned before, we modified lines 10-11, paragraph 1, Study design section, to provide greater clarity about the visits, in the following way:

“Treatment was provided on site at the following visits: baseline and weeks 4, 8, 12, 24, 36 y 48.” 

Acronyms need to be spelled out (ex, PDVF, SAE). This is broadly focused journal with clinical and non-clinical readers, and these are not universally known acronyms, which limits the accessibility of the paper.

Response: We have revised all acronyms to verify that they are spelled out at first mention.

Discussion: As noted above, results are interpretated treating study retention as the same as HIV care retention. This needs to be addressed.

Response: We have adjusted the Discussion, and particularly, paragraph 2 so that it is consistent with what was actually measured, that is, retention in the study. We have removed any interpretation of these results as “retention in HIV care”.

It would be helpful to situate these findings more explicitly in the context of current treatment practice.

Response: As previously mentioned, in the Conclusion section, paragraph 1, lines 4-5, we situate these findings in the current treatment practice:

“Thus, results support the use of DTG-based treatments in this population, as it is recommended by the WHO [19] in naïve patients in general.”

Given the design, I do not consider the characteristics of the site to be a limitation. It seems that the argument of the paper is that integrated approaches are needed and this study is providing evidence to support that. Therefore, I would not call the site a limitation.

Response: We thank the reviewer for noting this and from reframing the characteristics of the site from a more interesting and enriching perspective. We have removed the reference to the site characteristics from the limitations paragraph. We have relocated that fragment in the Discussion section, paragraph 7, lines 8-10, and in the Conclusion section, paragraph 2, lines 5-8, where we consider it will better support the argument of this manuscript.

In order to fully appreciate the discussion of an integrated approach in the discussion, as noted above, we need to know more about what services were provided and how they were delivered.

Response: We completely agree with the reviewer’s comment. We have added a detailed description of the characteristics and components of our trans-affirmative healthcare service, as we already explained in a previous response. 

Reviewer #2 

This is a well written observational study of ART-naïve TGW who are initiated on ART by an experienced clinical trial team – sensitive to trans-challenges. I have really very little to remark.

Response: We thank the reviewer for the positive comments.

Why do authors call this a pilot study? Will they propose a large efficacy trial? I didn’t see the arguments for this in the discussion.

Response: We agree with the reviewer’s comment. This study was not followed by a larger trial, later. Therefore, we removed the term “pilot” in reference to this study. 

In lines 280-286 – authors present data on retention from other studies and show that retention in the present study is above that of other studies reported. I would have expected nothing less from an experienced trial team – supported by Trans peer mobilisors. In fact, in the title of the manuscript – “retention in care” is emphasized. I wonder if retention is the main study outcome. Isn’t the study a description of care cascade indicators? I suggest a title change, replacing “retention in care” with “Care cascade indicators” in…..

Response: We thank the reviewer for this suggestion. However, we are not completely sure that “care cascade indicators” would be the accurate term to include in the title. Generally, studies on the care cascade start with the all the diagnosed people in the general population, and then, calculate the proportion of those who initiated treatment, and so on. In contrast, we started with a convenience sample of 61 participants (i.e., selected intentionally and recruited by us), all of whom initiated ART as part of the study. These characteristics reduce the possibility of comparison of our results with those of studies on the care cascade indicators.

Reviewer #3 

The analyses of this trial are pretty straightforward and I don't see any changes needed. Though, given the longitudinal data collection performed on this participants, it is disappointing that the authors did not perform and present longitudinal analyses of this cohort. Potentially a survival analysis would be interesting to determine how long participants were retained.

Response: As recommended by the reviewer, we have conducted and included a survival analysis. The analysis conducted is described in the Statistical analysis section, paragraph 2, lines 10-11, in the following fragment:

“A survival analysis of retained participants was performed with R (survival package). Time to event was calculated by Kaplan-Meier method.”

We describe the results of this analysis in the Results section, in the following fragment:

“The starting point for this analysis is baseline visit when participants received the first treatment medication. Time between this visit and the next medication dispensed (next visit date) was calculated until the first event of no medication dispensed occurred. When this happened, the participant was considered “not retained in the study” for that time point. The survival analysis performed shows that the probability of retention in the study decreases with time (Fig 2). In the first month after enrollment (week 4), the mean retention probability is 92%. By month 2 (week 8) it descends to 87%; in month 3 (week 12) it descends to 81%; and in month 6 (week 24) is 73%. The mean retention probability by month 9 (week 36) is 68% and by the end of the study (week 48) is 65%. The greater decrease in retention probability occurred in the first to third month of treatment.”

These results are also shown in Figure 2.

The more problematic parts of this study are related to the study design. The authors mention in lines 326-327 that this single-arm trial will not allow for conclusions about efficacy and that the study lacks generalizability (lines 344-345). Though it struck me that the first half of the discussion (approximately lines 273-299) were about the efficacy of the trial and comparing the results to other results. While I think it's important to compare the results with other studies, it behooves to provide some context as to why these are different. The retention and viral suppression results could be due to the intervention, but could also be due to differences in the population. For instance, take this statement (lines 284-285):

Moreover, the results of the current study showed better retention than the estimate found in a retrospective study conducted by our research group in a public hospital of Buenos Aires, reporting a retention of 46.4% at 1 year of treatment initiation [25].

As the authors note, the differences could be due to differences in the first line treatment, but could also be due to differences in the population. The authors should provide more information on where these results are coming from and why they might be different. For instance, the authors found an association with age in their data; what were the age distributions in other studies? Maybe worse outcomes would be expected in those studies given the age distributions.

Response: We agree with the reviewer’s remark. We have removed the conclusions about the efficacy of the treatment, as they contradicted the limitation of a single-arm trial. However, comparisons with other studies were conducted in order to have external references or criteria to appraise the retention rate obtained in this study. There is no intention to generalize results, as the sampling method does not allow it.

Regarding these comparisons, we have reviewed the studies that are referenced in the Discussion. We found no significant differences between the characteristics of their sample and ours. That led us to conclude that differences in retention and viral suppression are not probably due to characteristics of the populations, including age, and may be more probably related to the treatment and the characteristics of our HIV care service (trans-affirmative). We have added a fragment in the Discussion section, paragraph 2, to make these possible explanations clearer:

It is worth noting that when these previous studies were conducted, ART combinations did not include DTG as the first line, preferred treatment. It is possible that the study regimen exhibited a better tolerability than the standard first-line at the time of designing this protocol. Additionally, unlike these previous studies, in this trial, treatment was provided in the context of a trans-affirmative care service. This characteristic may have contributed to increased retention in the study.

Finally, given this is a pilot or proof of concept study, I think the authors should provide more information on how these results should be used. Are further studies warranted? If so, how should they be designed? Who should be the target population?

Response: We appreciate the reviewer´s suggestion. Firstly, we have removed the term “pilot” from the description of this study. This study was finally not followed by a larger trial, later. The reason is that, since this study was designed and conducted, dolutegravir-based regimens have become the first line treatment for naïve patients. We mention this in the Discussion section, paragraph 5, lines 2-4: “This study supports the current WHO recommendations regarding the use of DTG-based therapies and, in line with other studies [32-33-34-35] this regimen showed excellent safety and good tolerance with adherence percentages of 95%.”

We highlight this in the Conclusion section, paragraph 1, lines 4-5: “Thus, results support the use of DTG-based treatments in this population, as it is recommended by the WHO [19] in naïve patients in general.” Therefore, the efficacy, safety and tolerance of these regimens in this population and other are well-established.

However, we do recommend that future studies continue to explore the impact on retention and other HIV-related outcomes of a more comprehensive, trans-affirmative approach in healthcare services for transgender people and the feasibility of its implementation. We have added this recommendation in the Conclusion section, paragraph 2, lines 5-8:

“As these results suggest, a trans-affirmative approach in care and the inclusion of peer navigators may facilitate access to healthcare and favor greater retention. Thus, it is recommended to continue exploring the feasibility of the implementation of these strategies in public health care services.”

Specific comments:

1. (line 174) I am not sure what "independent samples" means in this statement.

Response: We agree with the reviewer’s remark. We have removed “independent samples”, as it was confusing.

2. (line 300) "The more significantly characteristic" does not make sense.

Response: We thank the reviewer for noting this. We have corrected that sentence.

---

## [Decision Letter · Decision Letter 1]

19 Dec 2022

Retention among transgender women treated with Dolutegravir associated with Tenofovir/Lamivudine or Emtricitabine in Argentina: TransViiV study

PONE-D-22-15918R1

Dear Dr. Frola,

We’re pleased to inform you that your manuscript has been judged scientifically suitable for publication and will be formally accepted for publication once it meets all outstanding technical requirements.

Kind regards,

Patricia Evelyn Fast, MD, Ph.D.

Academic Editor

PLOS ONE

Additional Editor Comments (optional):

Please do carefully consider the suggestion from Reviewer #1 to add a proportion to the abstract, which will further strengthen the paper.

Reviewers' comments:

Reviewer's Responses to Questions

**Comments to the Author**

1. If the authors have adequately addressed your comments raised in a previous round of review and you feel that this manuscript is now acceptable for publication, you may indicate that here to bypass the “Comments to the Author” section, enter your conflict of interest statement in the “Confidential to Editor” section, and submit your "Accept" recommendation.

Reviewer #1: (No Response)

Reviewer #2: All comments have been addressed

Reviewer #3: All comments have been addressed

2. Is the manuscript technically sound, and do the data support the conclusions?

Reviewer #1: Yes

Reviewer #2: Yes

Reviewer #3: (No Response)

3. Has the statistical analysis been performed appropriately and rigorously? 

Reviewer #1: Yes

Reviewer #2: Yes

Reviewer #3: (No Response)

4. Have the authors made all data underlying the findings in their manuscript fully available?

Reviewer #1: Yes

Reviewer #2: Yes

Reviewer #3: (No Response)

5. Is the manuscript presented in an intelligible fashion and written in standard English?

Reviewer #1: Yes

Reviewer #2: Yes

Reviewer #3: (No Response)

6. Review Comments to the Author

Reviewer #1: The authors have provided a comprehensive revision and thoughtful responses to suggestions and questions. While they added recognition of gaps in HIV care and treatment outcomes, in addition to prevalence, they do not include any proportions to support their statement:

"In Argentina, transgender women (TGW) have a high HIV prevalence (34%). However, this population shows lower levels of adherence, retention in HIV care and viral suppression than cisgender patients".

Values are needed to support this second statement in the abstract; this content should also be added to the introduction of the paper.

Reviewer #2: I am satisfied with authors' responses and believe this manuscript makes a useful contribution.

Reviewer #3: (No Response)

7. PLOS authors have the option to publish the peer review history of their article (what does this mean?). If published, this will include your full peer review and any attached files.

Reviewer #1: No

Reviewer #2: No

Reviewer #3: No

---

## [Editor Report · Acceptance letter]

10 Jan 2023

PONE-D-22-15918R1 

Retention among transgender women treated with Dolutegravir associated with Tenofovir/Lamivudine or Emtricitabine in Argentina: TransViiV study 

Dear Dr. Frola:

I'm pleased to inform you that your manuscript has been deemed suitable for publication in PLOS ONE. Congratulations! Your manuscript is now with our production department. 

Kind regards, 

on behalf of

Dr. Patricia Evelyn Fast 

Academic Editor

PLOS ONE